# Deep Learning for Improved Subsurface Imaging: Enhancing GPR Clutter Removal Performance Using Contextual Feature Fusion and Enhanced Spatial Attention

Yi Li [1], Pengfei Dang [1,2,*], Xiaohu Xu [3] and Jianwei Lei [4]

1    School of Civil Engineering, Guangzhou University, Guangzhou 510006, China; 339355@gzhu.edu.cn
2    Earth System Science Programme, Faculty of Science, The Chinese University of Hong Kong, Shatin, Hong Kong 999077, China
3    Gold Leaf Production and Mamufacturing Center, China Tobacco Henan Industrial Co., Ltd., Zhengzhou 450000, China; 2112016181@e.gzhu.edu.cn
4    School of Water Conservancy and Civil Engineering, Zhengzhou University, Zhengzhou 450001, China; leijianwei0203@zzu.edu.cn
*    Correspondence: iempengfeid@gzhu.edu.cn; Tel.: +86-13609205349

**Abstract:** In engineering practice, ground penetrating radar (GPR) records are often hindered by clutter resulting from uneven underground media distribution, affecting target signal characteristics and precise positioning. To address this issue, we propose a method combining deep learning preprocessing and reverse time migration (RTM) imaging. Our preprocessing approach introduces a novel deep learning framework for GPR clutter, enhancing the network's feature-capture capability for target signals through the integration of a contextual feature fusion module (CFFM) and an enhanced spatial attention module (ESAM). The superiority and effectiveness of our algorithm are demonstrated by RTM imaging comparisons using synthetic and laboratory data. The processing of actual road data further confirms the algorithm's significant potential for practical engineering applications.

**Keywords:** ground-penetrating radar (GPR); contextual feature fusion module (CFFM); enhanced spatial attention module (ESAM); clutter removal; reverse time migration (RTM)

## 1. Introduction

Ground-penetrating radar (GPR) is a non-destructive detection method that involves transmitting and receiving high-frequency electromagnetic waves under the ground through the transmitting antenna. When propagating in the underground media, as electromagnetic waves encounter electrically different interfaces, they will be reflected. The spatial location, structure, form and burial depth of ab underground anomaly can be inferred from the received electromagnetic waveform, amplitude intensity and time changes [1]. Due to the characteristics of non-destructive and high-resolution imaging [2,3], this technology has been successfully applied in agricultural detection [4], building-disease detection [5–7], pavement detection [8–10], glacier investigation [11], archeology [12] and other fields. In the actual detection work, the complex environment of the work area and the inhomogeneous distribution of the subsurface medium make the collected data subject to clutter interference, leading to accuracy degradation [13,14], causing difficulties in subsequent imaging and interpretation, and even causing misjudgment of the target anomalies [15]. Therefore, clutter removal is a necessary step in the pre-processing of GPR recordings.

Common clutter-removal methods are usually based on subspace or sparse representation [16,17]. To adaptively implement clutter removal from field GPR recordings [18], the advantages of multiple signal processing techniques are utilized to combine independent component analysis (ICA) and principal component analysis (PCA) as a unique algorithm for removing GPR clutter and extracting the target signal [19]. To further remove the contamination of the target signal by clutter and noise, the GPR recording is divided into

different subcomponents using PCA, and the Gaussian curvature decomposition (GCD) method is applied to the PCA domain subspace to achieve the removal of complex random noise [20]. In research, the low-rank and sparse decomposition (LRSD)-based method has been shown to outperform the conventional methods in GPR clutter suppression [21,22]. A detailed comparison of robust principal component analysis (RPCA), morphological component analysis (MCA), and robust nonnegative matrix factorization (RNMF)-based clutter-removal methods in [23] demonstrates the superiority of the RNMF method. Similar to the RPCA method, a new clutter-removal method based on tensor-robust principal component analysis (TRPCA) was proposed to limit the sparsity of the target and reduce the complexity of the algorithm using different cost functions [15]. To exploit the multi-resolution and orientation information of GPR images, the TRPCA-based bandpass filter algorithm was proposed to obtain superior background noise removal [24]. With the continuous research in data analysis, the sparse representation-based clutter-removal method was proposed, which utilizes a random dictionary and sparsely represents the GPR signal, thereby successfully suppressing the clutter interference and proving the effectiveness of the improved algorithm [25,26].

In recent years, with the research development of deep learning algorithms, they have been widely used in the field of GPR clutter removal because of their high efficiency and end-to-end advantages, while significantly outperforming traditional algorithms in terms of accuracy [27,28]. Based on the traditional algorithm, a robust auto encoder (RAE) clutter suppression method was proposed in combination with a deep learning framework, and the experimental results prove that the algorithm outperforms the current state-of-the-art clutter removal algorithms [29,30]. Feng et al. [31] proposed a framework of deep convolutional denoising autoencoders (DAE) with network-structure optimization, which has fidelity in GPR clutter removal. In order to improve the accuracy of the deep learning framework for GPR-data clutter removal, scholars have made a series of improvements to the deep learning framework. For example, the application of residual dense blocks significantly improves the generality of the network [32], and the improvement of the network structure not only provides excellent performance in removing non-smooth stochastic noise and clutter, but also effectively protects the edge information of GPR recording and obtains higher network performance [33,34].

The data preprocessing of GPR ensures the readability of the data and lays the foundation for the subsequent data processing. In order to enable further analysis of GPR data, the necessary imaging processing of GPR data is required. Migration imaging of GPR records can accurately focus the contour information and location information of the target anomaly, thereby facilitating the analysis of the distribution state of subsurface anomalies, and the quality of imaging directly depends on the quality of the processed GPR recordings [35]. To improve the quality of migration imaging, Feng et al. [36] proposed a migration-imaging method based on accurate velocity estimation and total variation (TV) denoising, and experiments showed that the method suppressed the effects of artifacts such as noise interference, multiple interference, arc clutter, and crosstalk, and improved the quality and accuracy of migration results. Clutter in GPR data seriously interferes with the imaging quality of the data, and jin et al. proposed the use of a 2D wavelet transform (WT) and F-K migration to identify fractured rocks, and the proposed method can identify fractured rock regions [37]. Combining deep learning algorithms with traditional algorithms enables high accuracy imaging of GPR data, and the YOLOv3 model has been used to identify the subsurface pipeline region in GPR data and convert it into a binary image by migration focusing the hyperbolic response of the pipeline to estimate the horizontal location and burial depth of the pipeline [38].

Table 1 presents a summary of frequently used algorithms along with their respective advantages and disadvantages.

**Table 1.** Progress in GPR clutter-removal methods.

| Methods | Category | Typical Case | Pros and Cons |
|---|---|---|---|
| Traditional Algorithm | Subspace | ICA, PCA, RPCA, MCA, RNMF, Tensor RPCA, TRPCA-BPF | Low computational cost and simple method; Requires manually given parameters, Difficult to handle complex situations. |
| | Sparse Representation | LRSD, K-SVD, Dictionary Learning | Denoising performance is more stable; requires manually given parameters, higher computational cost |
| Deep Learning Algorithm | - | Autoencoder, CR-Net, Declutter-GAN | Superior denoising effect; Difficulty in accurately capturing target signals |

This paper addresses the issue of ground penetrating radar (GPR) records being frequently disrupted by clutter caused by the uneven distribution of subsurface media, which impedes the delineation and accurate positioning of anomalies. To tackle this challenge, we initially proposed a deep learning-based clutter removal scheme for GPR records. The traditional deep learning network Res-UNet exhibits limited ability in capturing the signal features of GPR data. As a solution, we introduced the integration of CFFM and ESAM to enhance the deep learning network's ability to capture GPR signals' features, particularly for those weak signals submerged in clutter. This enhancement significantly improves the separation of weak signals, and ultimately results in a comprehensive workflow for accurately locating underground anomalies using GPR records.

By combining deep learning preprocessing with RTM imaging, we enhance the imaging quality of RTM and effectively utilize GPR data for pinpointing subsurface anomalies, providing guidance for practical production work. This demonstrates the potential of the method presented in this paper.

The organizational structure of this article is as follows. First, we discuss the causes and solutions of clutter in GPR records and acknowledge that while Res-UNet performs well in removing clutter from GPR data, its ability to capture signal features of GPR data remains insufficient. We then propose the CFFM and ESAM to enhance this deep learning network's ability to capture signal features. Using synthetic and experimental data, we detail the superiority of our proposed algorithm compared to traditional algorithms and conventional deep learning methods. Furthermore, we demonstrate the algorithm's practical applicability and potential to assist actual production work through the analysis of measured road data.

## 2. Materials and Methods

### 2.1. DCT Dictionary Learning

The discrete cosine transform (DCT) dictionary has a superior ability to decompose periodic signals. For a two-dimensional (2D) signal $Y$ of size $M \times N$, its 2-D DCT transformation can be written as

$$F(u,v) = c(u)c(v) \sum_{i=0}^{M-1} \sum_{j=0}^{N-1} Y(i,j) \times \cos\left[\frac{(2i+1)\pi}{2M}\right] \cos\left[\frac{(2j+1)\pi}{2N}\right] \qquad (1)$$

where $u = 0, 1, 2, ..., M - 1$; $v = 0, 1, 2, ..., N - 1$; $F(u, v)$ denotes the DCT coefficient; $c(u)$ and $c(v)$ denote the compensation coefficients and are defined as

$$c(u) = \begin{cases} \sqrt{\frac{1}{M}}, & u = 0 \\ \sqrt{\frac{2}{M}}, & u \neq 0 \end{cases} \tag{2}$$

$$c(v) = \begin{cases} \sqrt{\frac{1}{N}}, & v = 0 \\ \sqrt{\frac{2}{N}}, & v \neq 0 \end{cases} \tag{3}$$

For the complete dictionary obtained after the DCT transformation, a new overcomplete dictionary is obtained by finer frequency sampling [17], and the DCT overcomplete dictionary constructed in this paper is shown in Figure 1.

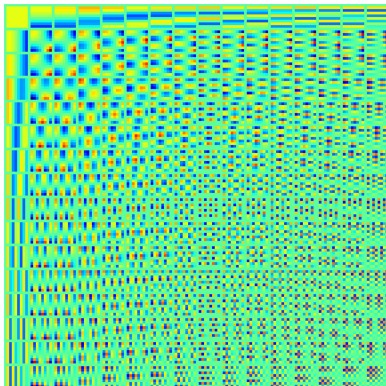

**Figure 1.** DCT over-complete dictionary.

*2.2. Contextual Feature Fusion Module*

CFFM utilizes the property of dilated convolution to improve semantic segmentation performance [39]. Dilated convolution has been shown to extract more detailed feature information for the target task, and we exploited this property to design our module. The module is designed to ensure that the network can better capture the detailed information of the target signal, allowing for the separation of valid and spurious signals. On the one hand, dilated convolution allows for the expansion of the perceptual field without losing resolution or increasing the number of parameters, enabling the detection of a larger range and obtaining accurate location information. On the other hand, by setting different expansion rates, multi-scale feature information is obtained, which can reduce the dependence on contextual information. The feature fusion module is formed by dilated convolution to generate multi-scale and multi-resolution feature maps. Finally, the feature maps of different scales are united by the concatenate operation, resulting in feature maps that contain both rich semantic information and local detailed information recorded by GPR. This improves the capture of detailed information, as shown in the specific structure illustrated in Figure 2.

Dilated convolution, a specialized convolution operation, as shown in Figure 3, effectively expands the receptive field by incorporating holes into the convolution kernel. This approach enhances the receptive field without increasing the convolution kernel's size or stride, providing benefits such as maintaining parameter count and computational complexity. By inserting spaced hole points within the convolution kernel, dilated convolution facilitates the sampling of input data, ultimately achieving receptive-field expansion. As this method refrains from increasing kernel size or stride, it avoids escalation in the numbers of parameters and computations while still improving the model's performance.

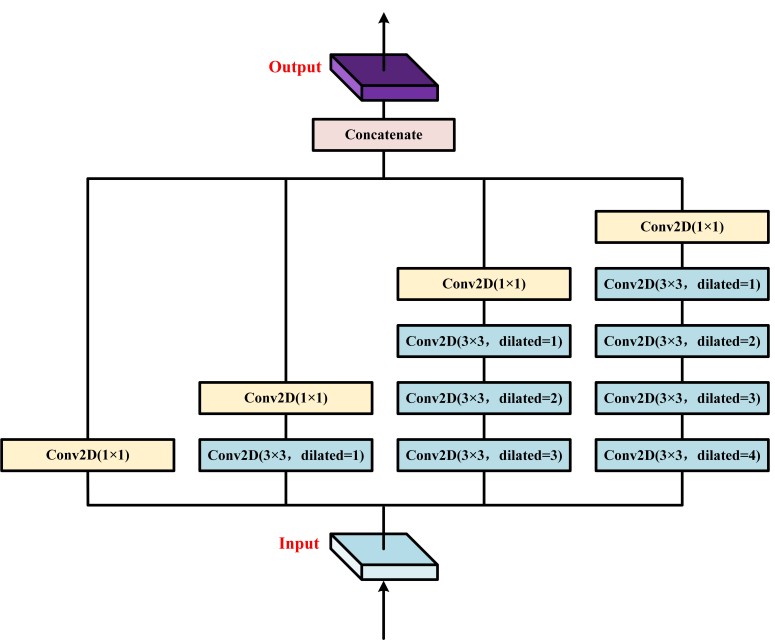

**Figure 2.** Contextual feature fusion module.

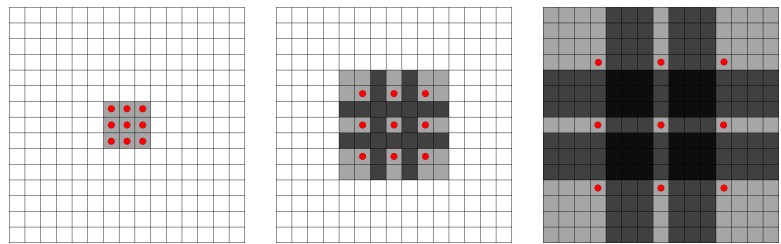

**Figure 3.** Dilated convolution.

### *2.3. Enhanced Spatial Attention Module*

The attention mechanism is a powerful tool for improving the performances of deep learning networks, as it mimics the biological process of focusing on relevant information and suppressing irrelevant information [40]. In digital image segmentation tasks, attention mechanisms have been used to achieve superior performance. These mechanisms can be easily integrated into existing deep learning frameworks, leveraging the relationships between spatial regions and channels to highlight relevant features. One specific example of an attention mechanism is the attention gate (AG) module, which is based on the U-Net framework. The AG model focuses on salient feature shapes and sizes, utilizing multi-scale information to improve prediction performance on different datasets and training scales. Additionally, it maintains computational efficiency. Another example is the squeeze and excite network (SE-Net), which uses SE blocks to recalibrate relevant channel feature maps and ignore irrelevant features [41]. These SE blocks have been shown to offer significant performance improvements with a minimal additional computational cost, making them a state-of-the-art technique in deep learning architectures [42].

The proposed module utilizes scale transformation to enhance the detailed information of the target signal, and incorporates contextual information in GPR recordings to avoid information loss. A residual connection is also utilized to reinject upstream information into downstream operations. The output feature map is obtained by multiplying the original information with the attentional feature map. The specific structure of the enhanced spatial attention module is illustrated in Figure 4.

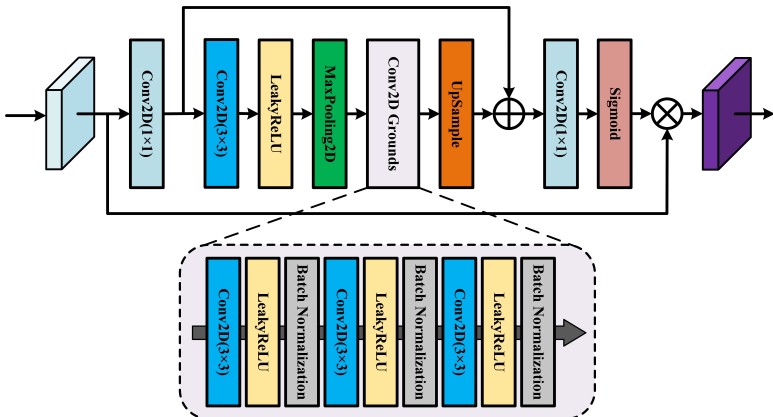

**Figure 4.** Enhanced spatial attention module.

### 2.4. Hybrid Loss Function

In deep learning, the loss function plays a crucial role in comparing the network output to real data, determining the gradient, and updating the network weights. The selection of an appropriate loss function can greatly impact the performance of the network [43]. Mean squared error (MSE) is widely used, but it can lead to over-smoothing and data distortion. To address this issue, this paper proposes a hybrid loss function that combines MSE and Laplace pyramid loss functions. The MSE component measures the error between data pixels while the Laplacian Pyramid component captures error at different scales, emphasizing the detailed information in GPR data. This hybrid approach aims to improve the overall performance of the network [44,45].

The expression of the MSE loss function is given by:

$$MSE(x,y) = \frac{1}{N} \sum_{i=1}^{N} (x_i - y_i)^2 \tag{4}$$

where $x$ denotes the real data, $y$ denotes the predicted data, and $N$ denotes the total number of matrix elements.

The expression of the Laplace pyramid loss function is:

$$Lap(x,y) = \frac{1}{N} \sum_{i=1}^{N} \sum_{j=1}^{M} |L^j(x_i) - L^j(y_i)|_1 \tag{5}$$

where $x$ denotes the real data; $y$ denotes the predicted data; $L^j(x_i)$ and $L^j(y_i)$ denote the $j$th level of the Laplace operator for data $x$ and $y$, respectively; and $M$ equals four in the training process.

Thus, the hybrid loss function expression is obtained as:

$$Loss(x,y) = MSE(x,y) + Lap(x,y) \tag{6}$$

The hybrid loss function can not only guarantee the data quality of the predicted GPR recordings, but also capture more texture and structural information in the GPR recordings, while capturing more detailed information to ensure better results of the reconstructed GPR recordings.

### 2.5. Structure of the Deep Learning Network

This paper presents an improved version of the traditional residual U-Net (Res-UNet) network for processing GPR data. The proposed network, referred to as CFFM-ESAM-Res-UNet, incorporates two novel modules—the CFFM and the ESAM—to enhance the ability of network to capture semantic and local detail information of GPR data. The CFFM module leverages the multi-resolution property of GPR data to extract the contextual feature maps,

and the ESAM is used to selectively enhance the target regions of interest while suppressing the irrelevant background regions. Figure 5a illustrates the traditional UNet framework, and Figure 5b illustrates the CFFM-ESAM-Res-UNet framework proposed in this paper.

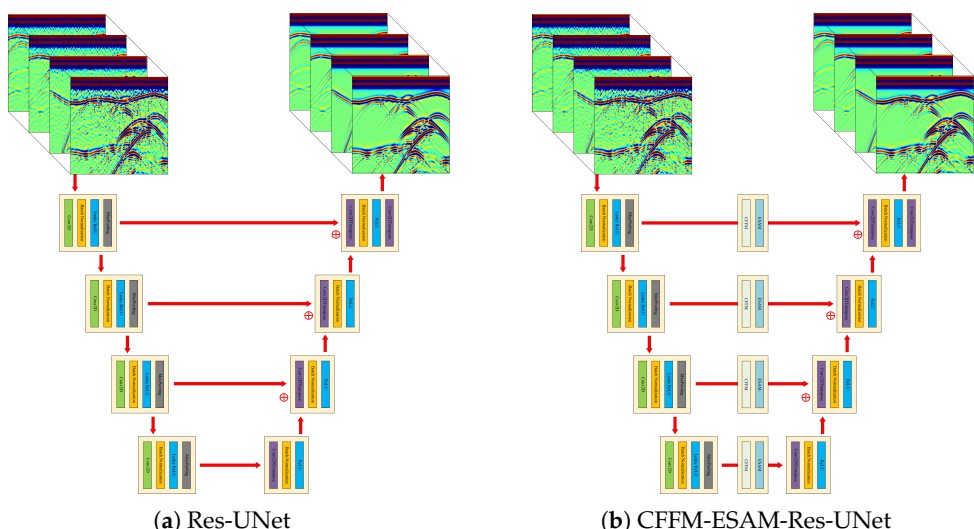

(**a**) Res-UNet         (**b**) CFFM-ESAM-Res-UNet

**Figure 5.** The structure of the deep learning network.

The proposed network was trained using a dataset of 43,217 samples, consisting of both synthetic data and real data. Eighty percent of the data were used for training, and the remaining 20% werer used for testing to evaluate the performance of network and prevent overfitting. The network was trained using the root-mean-square prop (RMSprop) optimization method for a total of 1000 epochs. The results of the training loss function for the proposed network and the traditional network are shown in Figure 6a and Figure 6b, respectively. Concurrently, to demonstrate the alterations in the network's prediction performance throughout the training process, we employed the mean peak signal-to-noise ratio (PSNR) value of the validation dataset as the evaluation metric for the network. The PSNR is computed using the following formula:

$$PSNR = 10 \cdot \log_{10}(\frac{MAX_I^2}{MSE}) \tag{7}$$

where $MAX_I$ is the maximum value of the data and $MSE$ is the mean square error of the data. Larger values indicate better image results.

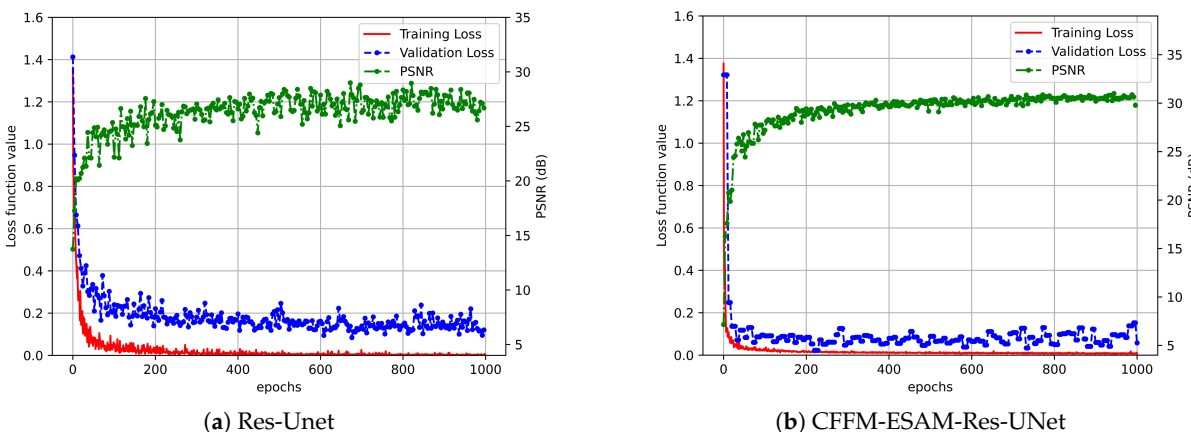

(**a**) Res-Unet         (**b**) CFFM-ESAM-Res-UNet

**Figure 6.** The value of the hybrid loss function during the network training.

The hybrid loss function values shown in Figure 6 demonstrate that the hybrid loss function oscillations decreased sharply in the first 200 epochs, which indicate that the training process was constantly optimized to make the predicted data correspond better to the real data, and the network training process has good convergence. After 200 epochs, the curve tends to smooth out, and the oscillations decrease and even become a straight line, which shows that the prediction output of the network has generally corresponded well to the real data, and the detailed part of the prediction output has been optimized on this basis. Comparing the loss functions in Figure 6a,b, it is observed that the traditional Res-UNet network exhibits stronger oscillations in its training process compared to the proposed CFFM-ESAM-Res-UNet network, with a worse convergence rate. Additionally, the CFFM-ESAM-Res-UNet network demonstrated more stable performance on the validation set and achieved smaller network losses, indicating that it has superior generalization capabilities. These results demonstrate the effectiveness of the proposed CFFM-ESAM-Res-UNet network in improving the overall performance of the network.

*2.6. Reverse Time Migration*

The principle of the RTM imaging technique is based on the fact that when the electromagnetic wave field propagates backwards along the time axis and eventually pushes the electromagnetic wave field back to zero, the energy of the reflected and diffracted waves in the GPR recordings will be converge to the real spatial position from where they were generated, followed by the use of corresponding imaging conditions to complete the overall RTM imaging process. Based on the principle of temporal consistency, the imaging conditions in the usual sense are expressed as:

$$I(x,z) = \sum_{m=1}^{M} \sum_{n=1}^{N} S_m(x,z,t_n) R_m(x,z,t_n) \tag{8}$$

where $I(x,z)$ denotes the RTM imaging result; $S_m(x,z,t_n)$ and $R_m(x,z,t_n)$ denote the forward and backward electromagnetic wave fields, respectively; $M$ is the number of traces of GPR recordings; $N$ is the total number of time steps; $x,z$ denote the spatial coordinates of the imaging results.

However, conventional intercorrelation imaging conditions, as in Equation (8), usually result in strong low-frequency noise in the shallow part in RTM imaging results, leading to the weak energy information in the deep part not being highlighted completely, so we adopted the normalized imaging condition at the source points, which not only can effectively suppress the appearance of low-frequency noise, but also can significantly complement the energy of the deep information and improve the imaging ability at the deep part. The modified imaging condition is:

$$I(x,z) = \frac{\sum_{m=1}^{M} \sum_{n=1}^{N} S_m(x,z,t_n) R_m(x,z,t_n)}{\sum_{m=1}^{M} \sum_{n=1}^{N} R_m(x,z,t_n) R_m(x,z,t_n)} \tag{9}$$

*2.7. Comprehensive Workflow*

In this subsection, we present a systematic and cohesive workflow that outlines the steps and processes involved in our study.

To accurately locate underground anomalies and mitigate clutter interference stemming from the uneven distribution of subsurface media, this study introduces a GPR-data preprocessing technique based on deep learning, in conjunction with the RTM imaging method for the precise positioning of subterranean anomalies. The methodology primarily encompasses two components: GPR-data preprocessing and GPR-data RTM imaging. Initially, the acquired GPR data are processed through the CFFM-ESAM-Res-UNet framework to eliminate interference caused by the irregular distribution of subsurface media. Subsequently, the RTM imaging method was employed to achieve precise positioning of

underground anomalies. A schematic representation of the comprehensive workflow is illustrated in Figure 7.

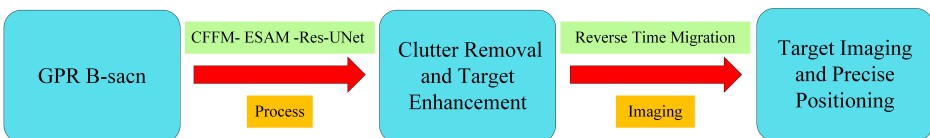

**Figure 7.** Comprehensive workflow for precisely identifying subterranean anomalies via GPR data analysis.

The overarching goal of this study is to establish a standardized process applicable to real-world GPR operations, ultimately offering guidance for practical engineering projects and enhancing overall productivity.

### 3. Results

*3.1. Synthesis of Data Test*

To demonstrate the necessity of clutter removal in GPR recordings and the superiority and effectiveness of the CFFM-ESAM-Res-UNet network, we used the finite-difference time-domain (FDTD) algorithm [46–49] to simulate a subsurface model which was composed of three layers: an air layer simulating the coupling between the antenna and the ground, a soil layer with a relative dielectric constant of 5, and a bedrock layer with a relative dielectric constant of 9. To generate a realistic scenario, we positioned a circular and an irregular cavity on the left side of the scene, and two empty pipes with distinct diameters and materials were situated on the right. The relative dielectric constants of the pipes walls were 9 and 4, respectively. An irregular, intersecting fracture was placed on the far right of the scene, further contributing to the complexity of the subsurface environment. The transmitting and receiving antennas were placed above the air layer, and the ricker wave signal with the main frequency of 400 MHz was used as the excitation source. The time step was set to 0.04 ns. There were a total of 1500 time steps and a time window length of 60 ns. Two-hundred and fifty pairs of excitation and reception points were set on the ground with uniform distribution. Using the B-scan transceiver method, the GPR record was obtained as shown in Figure 8b.

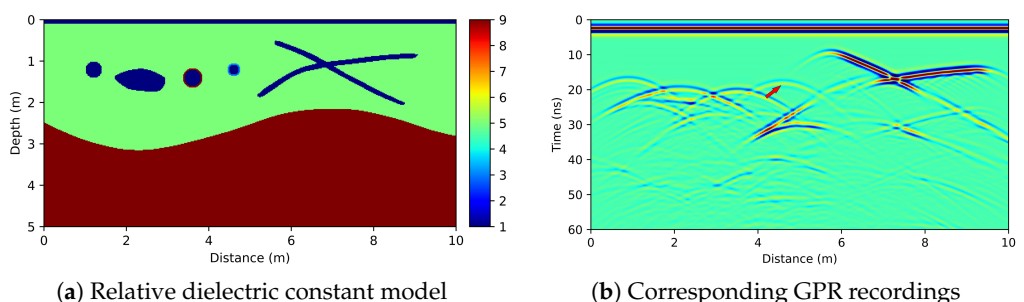

(**a**) Relative dielectric constant model　　　　(**b**) Corresponding GPR recordings

**Figure 8.** Synthesis model and corresponding GPR recordings.

From the GPR recording shown in Figure 8b, it can be seen that the reflected echoes of all anomalies correspond well with the anomalies set in the model. However, it is important to note that the subsurface medium is not always uniform, and often presents a stochastic distribution. Therefore, the model presented in Figure 8a must be modified to better match the actual subsurface situation. To address this issue, we propose the use of a stochastic medium model [50]. The modified model, shown in Figure 9a, considered the stochastic distribution of the subsurface medium. In this model, the average relative dielectric constant of the background medium was set to 5, with a variance of 0.2. The corresponding forward profile, shown in Figure 9b, demonstrated a closer match to the actual subsurface situation.

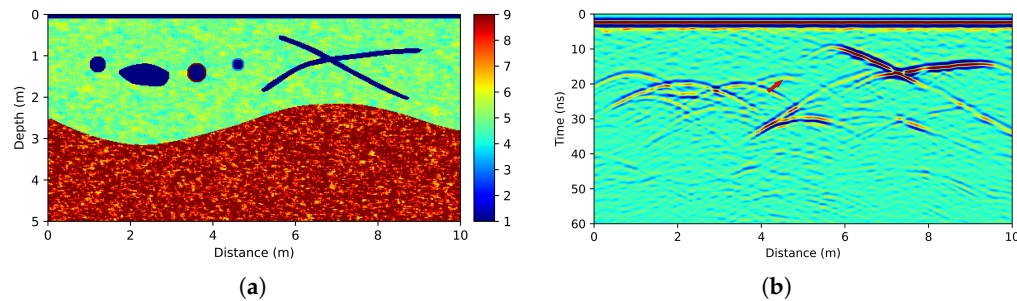

**Figure 9.** Synthesis of the stochastic media model and the corresponding GPR recordings. (**a**) Relative dielectric constant model with stochastic media; (**b**) corresponding GPR recordings.

The background in the GPR recording shown in Figure 9b is cluttered, and these cluttered backgrounds cause a certain degree of interference with the target reflection signal, which affects the subsequent data interpretation, making it prone to misinterpretation. The GPR profile with clutter shown in Figure 9b was processed with the conventional DCT dictionary learning, Res-UNet, and CFFM-ESAM-Res-UNet, and the processing results are shown in Figure 10.

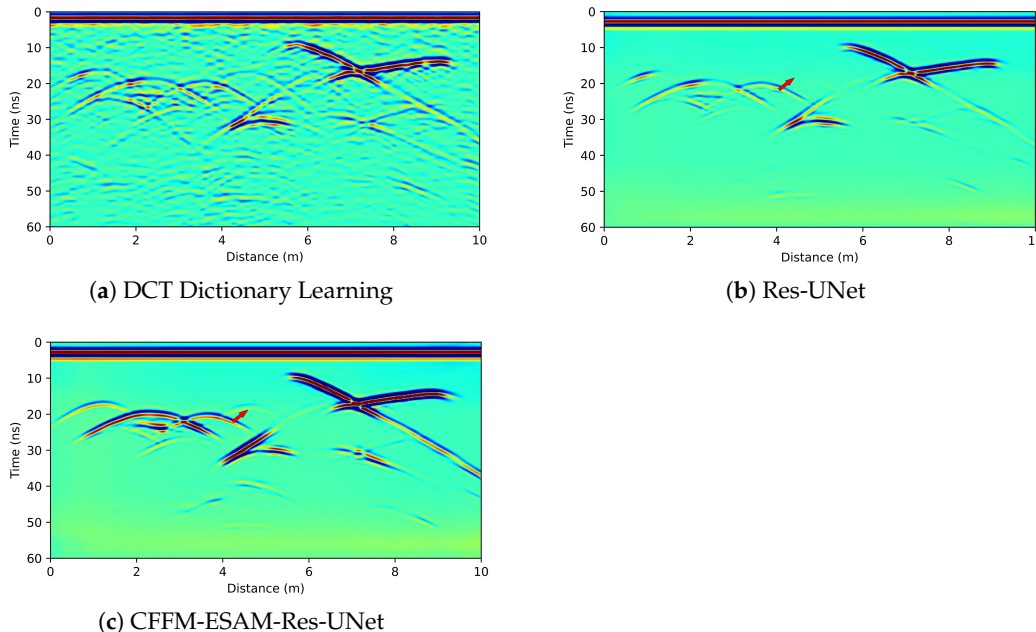

**Figure 10.** Synthetic-GPR-recording clutter-removal results for three methods.

By comparing the processing results in Figure 10, we found that while DCT dictionary learning can effectively reduce clutter in the profiles, it still leaves some clutter present, causing some degree of damage to the valid signal. Res-UNet removes the visible clutter, but the valid signal is still affected and not fully recovered. Owing to the CFFM framework's ability to capture an abundance of semantic information and the local details of GPR records, in conjunction with the attention mechanism inherent to the ESAM framework, CFFM-ESAM-Res-UNet effectively eliminates discernible clutter. Consequently, the retrieved target signals, particularly those exhibiting weak reflections, are thoroughly extracted and accentuated. To quantitatively evaluate the processing effect, we used the peak signal-to-noise ratio (PSNR) and structural similarity index measure (SSIM) algorithms as evaluation indexes for super-resolution results, as defined in Equations (7) and (10). The comparison results are presented in Table 2. It is crucial to note that synthetic data allowed us to perform quantitative analysis. This is attributed to the fact that, for synthetic models, we possess a

priori knowledge of the GPR data without clutter interference. Due to these considerations, we did not conduct a similar comparison with the measured data, as it is not feasible to obtain GPR data devoid of clutter interference for such data.

$$SSIM(x,y) = \frac{(2\mu_x\mu_y + C_1)(2\sigma_x\sigma_y + C_1)(\sigma_{xy} + C_3)}{(\mu_x^2\mu_y^2 + C_1)(\sigma_x^2\sigma_y^2 + C_2)(\sigma_x^2\sigma_y^2 + C_3)} \tag{10}$$

where $x$ and $y$ represent the comparison data (predict GPR data) and standard data (GPR data without clutter), respectively; $\mu_x$ and $\mu_y$ represent the means of data $x$ and $y$, respectively; $\sigma_x$ and $\sigma_y$ represent the variance of data $x$ and $y$, respectively; and $\sigma_{xy}$ represents the covariance of data $x$ and $y$. $C_1$, $C_2$, and $C_3$ are constants to avoid the denominator is zero. $C_1 = (K_1 \times L)^2$, $C_2 = (K_3 \times L)^2$, and $C_3 = C_2/2$ are usually taken. Generally, $K_1 = 0.01$, $K_2 = 0.03$, $L = 255$, and SSIM take the value range of [0, 1], the larger value means the image distortion is smaller.

As shown in Table 2, the computational results support the conclusions drawn from the visual analysis of the processing results in Figure 10. The DCT dictionary learning algorithm was found to be inadequate in completely removing clutter from the profile, leaving room for improvement in the processing effect. The traditional Res-UNet framework, while effective in clutter removal, causes a certain degree of distortion in the data. In contrast, the deep learning algorithm proposed in this paper achieved the highest PSNR and SSIM values, demonstrating its ability to effectively remove clutter while maintaining data fidelity. This is attributed to the improved framework of the proposed algorithm.

**Table 2.** Comparison of PSNR and SSIM after diferent methods processing.

| Method | PSNR (dB) | SSIM |
|---|---|---|
| GPR recording with clutter | 6.43 | 0.7449 |
| DCT Dictionary Learning | 15.57 | 0.8276 |
| Res-Unet | 28.40 | 0.9697 |
| CFFM-ESAM-Res-UNet | 31.65 | 0.9896 |

To illustrate the effect of clutter removal on the subsequent imaging of GPR data and to demonstrate the necessity of the clutter removal method. The profiles shown in Figures 8b, 9b and 10 were imaged with RTM, respectively, and the imaging results are shown in Figure 11.

The results depicted in Figure 11 demonstrate that clutter substantially impacts the quality of RTM imaging outcomes. When performing direct migration processing on data containing clutter and normalizing the clutter, the clutter interferes with anomalous signals or even obscures the target signal, leading to misinterpretation of the results. Additionally, it is evident that the DCT dictionary learning algorithm fails to fully eliminate the clutter within the silhouette, resulting in spurious anomalies that hinder the accurate interpretation of target body information. While the deep learning algorithm employing Res-UNet substantially removed clutter, it also partially compromised weak signals submerged within the clutter, causing a deficiency in the anomalous body positioning in the RTM imaging results and an inability to guarantee the fidelity of GPR data.

However, the CFFM and ESAM enhance the network's feature capture capability, enabling the extraction of weak signals submerged in clutter and achieving an separation between clutter and signal. Consequently, the position of the anomalous body in the subsequent RTM imaging is distinctly visible, ensuring the fidelity of the GPR data.

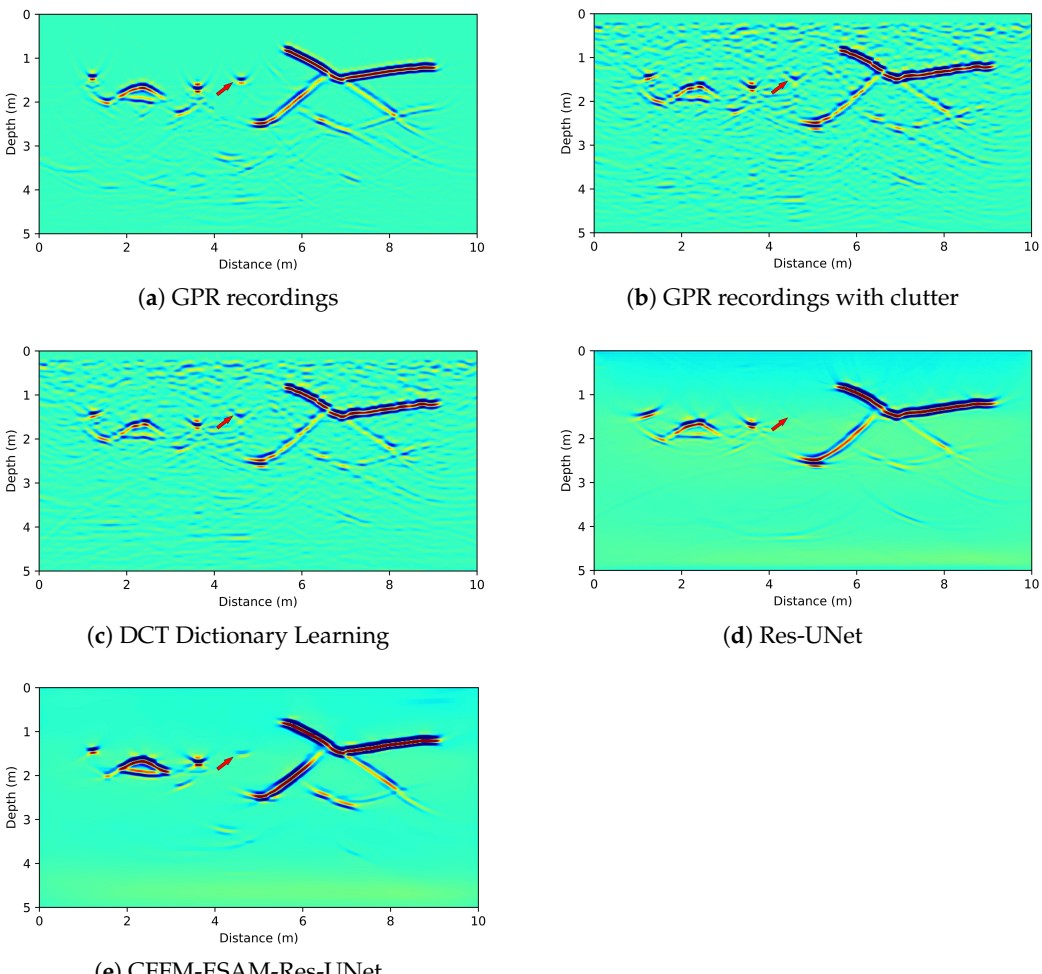

(**a**) GPR recordings

(**b**) GPR recordings with clutter

(**c**) DCT Dictionary Learning

(**d**) Res-UNet

(**e**) CFFM-ESAM-Res-UNet

**Figure 11.** RTM imaging results of synthetic GPR recordings after different processing methods.

### 3.2. Experimental Data

To verify the practicality and correctness of the proposed algorithm, we conducted an experimental study using three buried pipelines: two concrete pipelines and one metal pipeline. The GPR measurements were performed at a frequency of 400 MHz, with a time window length of 60 ns and B-Scan measurement mode. The diagram of physical laboratory model's distribution is shown in Figure 12a, and a total of 558 traces were collected at the acquisition site, as illustrated in Figure 13a. To further demonstrate the accuracy of the proposed algorithm, we recorded the materials, burial depths, and diameters of the pipes used in the experiment. The recorded data are presented in Table 3.

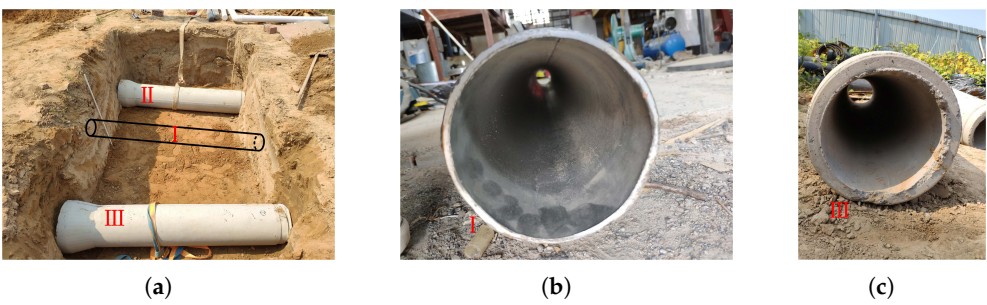

(**a**)

(**b**)

(**c**)

**Figure 12.** Diagram of experimental physical model distribution. (**a**) Physical model deployment of experimental data: (**b**) I, (**c**) III.

**Table 3.** Experimental physical models' parameters—data sheet.

| No. | Material | Depth (m) | Radius (m) |
|-----|----------|-----------|------------|
| I | Metal pipe | 0.10 | 0.20 |
| II | Concrete pipe | 0.80 | 0.90 |
| III | Concrete pipe | 0.90 | 0.90 |

From the collection site, the uneven distribution of dry and wet subsurface soils due to the weather affected the collected data. From the GPR recordings, the relative permittivity of anomalies II and III is closer to the background, resulting in low reflected echo energy and susceptibility to interference.

The clutter removal results shown in Figure 13b–d were obtained by using DCT dictionary learning, Res-UNet framework, and CFFM-ESAM-Res-UNet, respectively, for the original collected data.

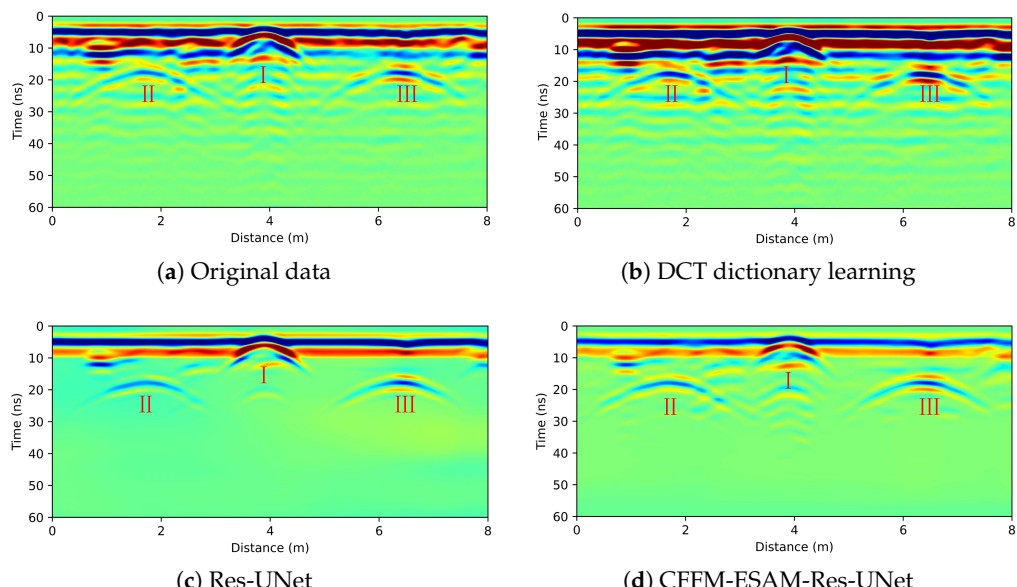

**Figure 13.** Experimental GPR recording Clutter removal results of different methods.

The results presented in Figure 13 reveal that in the original GPR record, the reflection signal is weak due to the similar relative permittivity values of the concrete pipe and the surrounding soil. Simultaneously, the uneven distribution of underground soil caused clutter interference, making it challenging to identify the reflection signal of the concrete pipe. Additionally, the metal pipe's proximity to the surface made it more susceptible to ground clutter interference. As a result, identifying the reflection signal of an anomaly in the original GPR record is difficult.

The experimental data processed by the DCT dictionary learning algorithm yielded unsatisfactory results, as clutter interference remained inadequately removed, making it difficult to identify effective reflection signals from the processed GPR records. Although Res-UNet removed a significant amount of clutter interference from the laboratory data, it also caused some damage to the effective reflection signal, which is primarily evident in the weak reflection anomaly of the concrete pipe. Notably, the leftmost abnormal reflected wave exhibits waveform distortion, weakening the energy of the reflected wave and potentially impacting subsequent precise positioning.

In contrast, the CFFM-ESAM-Res-UNet algorithm not only effectively removed clutter interference in the profile, but also retained high fidelity of the concrete pipe's echo signal. This is particularly true for weak-energy reflection signals, where the separation of clutter from the effective reflected signal achieved optimal results. The imaging outcomes depicted

in Figure 14, obtained by processing the GPR data with RTM, further substantiate this finding.

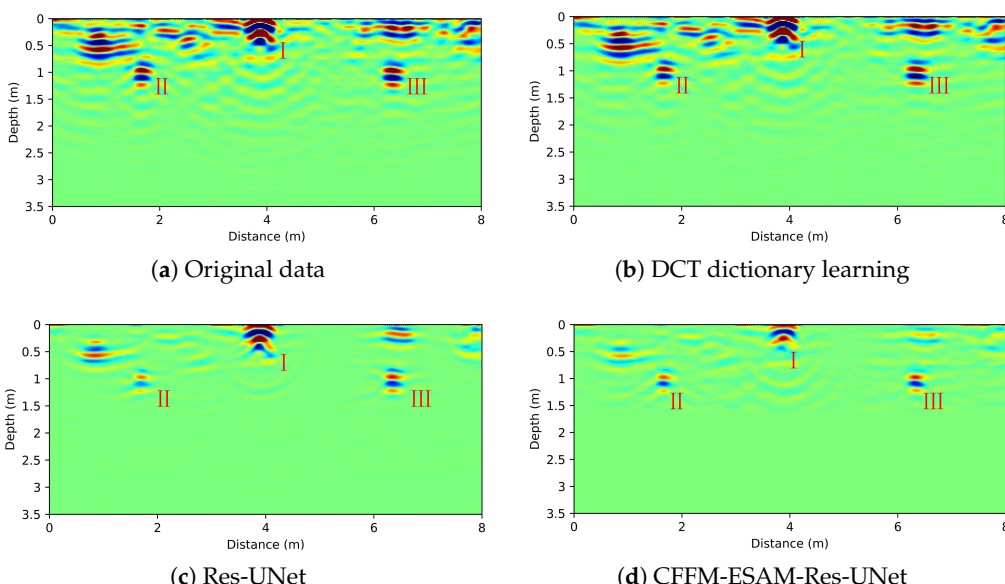

**Figure 14.** RTM imaging results of experimental GPR recordings after diferent methods processing.

The results displayed in Figure 14 support the same conclusion. In the original GPR record, the ground surface clutter interference energy is highly pronounced, rendering it nearly impossible to accurately pinpoint the precise positions of the left concrete pipe and the metal pipe in the RTM migration profile. Simultaneously, the RTM imaging results derived from data processed by the DCT dictionary also retained the majority of the clutter interference, making it difficult to accurately delineate the left concrete pipe and the metal pipe near the surface.

The RTM imaging results processed by Res-UNet reveal that the left concrete pipe's reflected wave energy was weak, and the waveform was distorted due to interference caused by data processing. Consequently, the final RTM imaging results exhibited energy weakening, fidelity crossing of GPR records, and strong surface reflection clutter, all of which contribute to the degradation of RTM imaging quality.

In contrast, CFFM-ESAM-Res-UNet attained the most optimal imaging effect. Not only was the clutter effectively removed, but the energy of the concrete pipe was also more concentrated due to the network's high fidelity. Simultaneously, the framework could distinguish surface strong reflection interference and exert a weakening effect, which is more conducive to delineating and accurately positioning anomalous bodies. By comparing the imaging results with the model's parameter data in Table 3, it is evident that the position parameters are in excellent agreement, corroborating the algorithm's correctness and the network's exceptional generalization capability.

### 3.3. Field Data

The algorithm was applied to GPR recordings of a road pavement, and the road was surveyed using an Impulse GPR with a 600 MHz central frequency. The location was in Zhengzhou University, Zhengzhou, Henan Province, China. The work site is shown in Figure 15, and the corresponding GPR recording are shown in Figure 16a. The profile consists of 938 channels of data with a recording duration of 40 ns. From the figure, it can be roughly judged that an anomaly exists at 20 ns, and other locations are difficult to interpret due to the presence of clutter. Based on the information from the collection site, the road surface is relatively flat and there is no collapsed area on the surface. Therefore, the clutter waves may have been caused by the uneven distribution of the subsurface medium.

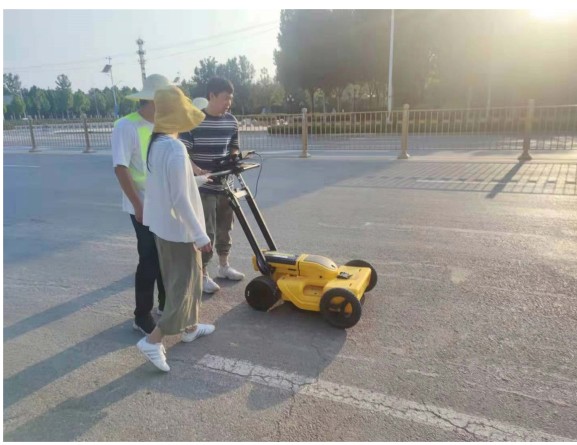

**Figure 15.** A photo of the collection of measurements.

The results shown in Figure 16b–d were obtained by using DCT dictionary learning, Res-UNet, and CFFM-ESAM-Res-UNet (separately) to remove clutter from the original acquisition data.

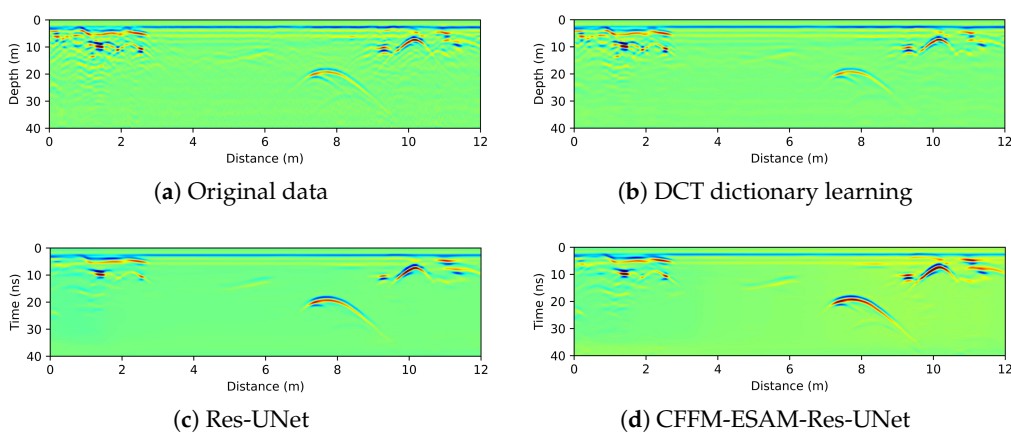

(**a**) Original data

(**b**) DCT dictionary learning

(**c**) Res-UNet

(**d**) CFFM-ESAM-Res-UNet

**Figure 16.** Clutter-removal results of different methods used on a real GPR recording.

The results shown in Figure 16 indicate that the DCT dictionary learning algorithm was less effective for the measured data. There is no significant difference between its results and the original profile, and it was still unable to interpret the anomalies. On the other hand, Res-UNet effectively removed the clutter information and highlightd the reflected echo signals of the anomalies, but some of the target signals were destroyed, and the data's fidelity cannot be guaranteed. In contrast, CFFM-ESAM-Res-UNet also achieved the removal of the clutter, and the effective signals were obviously highlighted, ensuring higher data fidelity. The GPR data in Figure 16 were processed with RTM, and the imaging results shown in Figure 17 were obtained.

The results shown in Figure 17 indicate that the DCT dictionary learning algorithm's results are not much different from the RTM results of the original data due to more clutter residues, and the cluttered anomaly distribution on the left side cannot be interpreted well. On the other hand, the processing results of Res-UNet can achieve the accurate localization of obvious anomalies; however, there may be excessive removal for the areas of waveform clutter. In contrast, the CFFM-ESAM-Res-UNet yielded good imaging results. Due to the fidelity of the network, the migration homing energy was more concentrated and the imaging results are more realistic and credible. By analyzing the imaging results, it can be preliminarily determined that there may be underground cavities in the shallow part of the left area, and there are two pipeline anomalies at the distances of 8 m, 9.5 and 10 m. There

is also a discontinuity of underground medium in the shallow layer at the position of 11 m, which is presumed to be a stratigraphic anomaly.

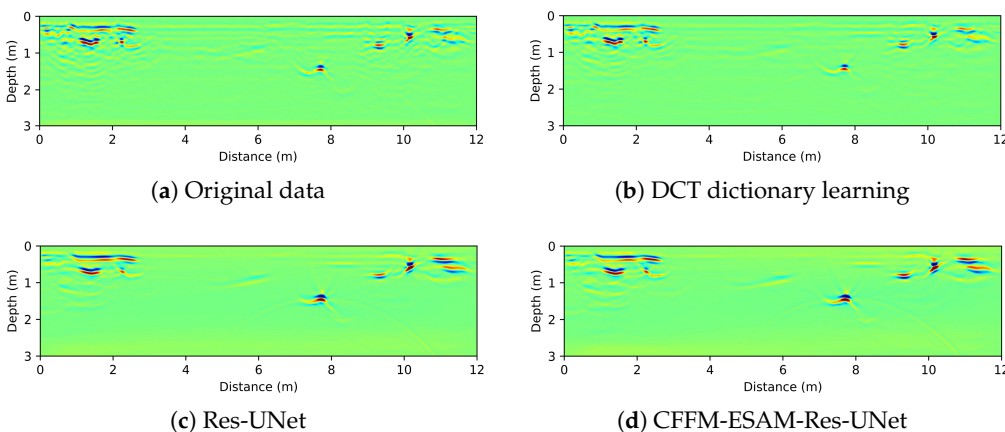

(**a**) Original data

(**b**) DCT dictionary learning

(**c**) Res-UNet

(**d**) CFFM-ESAM-Res-UNet

**Figure 17.** RTM imaging results of field GPR recordings after different processing methods.

## 4. Conclusions

The uneven distribution of subsurface media often leads to clutter interference in ground penetrating radar (GPR) data. This interference significantly impacts the hyperbolic shape and energy distribution of target signals, ultimately affecting the accurate localization of underground anomalies. To address this issue, we present a method that combines deep learning-based preprocessing and reverse time migration (RTM) imaging to enhance RTM imaging quality, thereby accurately localizing underground anomalies using GPR data. Our preprocessing method is a novel deep learning framework that is specifically designed to address clutter interference in GPR records, CFFM-ESAM-Res-UNet. This network builds upon Res-UNet. It integrates a contextual feature fusion module to expand the receptive field without sacrificing resolution or increasing parameter count, thereby extending the detection range and attaining precise positional information. Additionally, an enhanced spatial attention module is incorporated to emphasize valid signal details and suppress irrelevant information, thereby augmenting the network's optimization capability. This strategy bolsters the network's ability to capture target signal features.

We employed the DCT dictionary learning algorithm, Res-UNet, and CFFM-ESAM-Res-UNet to process synthetic data, illustrating the effectiveness of CFFM-ESAM-Res-UNet at handling GPR clutter interference and extracting weak signals submerged in clutter. The comparison of two quantitative indicators, PSNR and SSIM, further corroborates this conclusion. By conducting RTM on the processed profiles, the location information of weak signals is precisely determined, demonstrating CFFM-ESAM-Res-UNet's superiority in recovering weak signals and accurately localizing target anomalies.

We processed and performed RTM imaging on a set of laboratory data with known parameters. The results exhibit strong agreement with the actual location distribution, indicating the network's robust generalization capability. Concurrently, we utilized CFFM-ESAM-Res-UNet to process and analyze road GPR records obtained from practical engineering applications, mitigating the impact of uneven subsurface media distribution on the collected data and rendering the target more conspicuous. The underground anomaly distribution is accurately localized by RTM, providing valuable guidance for real-world production tasks.

In summary, the new deep learning framework CFFM-ESAM-Res-UNet has demonstrated its effectiveness in addressing clutter interference in GPR data and accurately localizing underground anomalies. Future development directions may include refining the network's architecture to further enhance its performance, expanding its applications to various subsurface environments, and exploring its potential for integration with other geophysical techniques to provide comprehensive and precise subsurface characterization.

**Author Contributions:** Conceptualization, Y.L., P.D. and J.L.; methodology, Y.L.; software, Y.L.; validation, P.D. and J.L.; formal analysis, Y.L.; investigation, X.X.; resources, J.L.; data curation, J.L.; writing—original draft preparation, Y.L.; writing—review and editing, P.D.; visualization, X.X.; supervision, J.L.; project administration, P.D.; funding acquisition, Y.L. All authors have read and agreed to the published version of the manuscript.

**Funding:** This research was funded by the Natural Science Foundation of Guangxi Province grant number 2018GXNSFAA294020, the Postdoctoral Office of Guangzhou City, China grant Number 62216246; the Natural National Science Foundation for Young Scientists of China, grant number 42204050; the Postdoctoral Office of Guangzhou City, China, grant number 62216242; and the Postdoctoral Program of International Training Program for Young Talents of Guangdong Province.

**Data Availability Statement:** Not applicable.

**Acknowledgments:** The authors would like to thank Hongyuan Fang, the School of Water Conservancy and Engineering, Zhengzhou University, for giving us the opportunity to collect the field data. They would also like to thank the support of TensorFlow 2.x for this research. In addition, they would also like to thank Tunitin for editing this article.

**Conflicts of Interest:** The authors declare no conflict of interest.

## Abbreviations

The following abbreviations are used in this manuscript:

| | |
|---|---|
| GPR | Ground-penetrating radar |
| CFFM | contextual feature fusion module |
| ESAM | enhanced spatial attention module |
| DCT | discrete cosine transform |
| TV | total variation |
| AG | attention gate |
| MSE | mean-square error |
| RMSprop | root mean square prop |
| FDTD | finite difference time domain |
| PSNR | peak signal-to-noise ratio |
| SSIM | structural similarity index measure |

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
