# Peer review of "Deep Learning for Improved Subsurface Imaging: Enhancing GPR Clutter Removal Performance Using Contextual Feature Fusion and Enhanced Spatial Attention"

_remotesensing, doi:10.3390/rs15071729_

Round 1

Reviewer 1 Report

The research addresses the removal of clutters and improvement of ground penetrating radar images by up-to-date deep learning architectures. The motivation and methodology are clear. However, there are some problems in the interpretation of the results (Q.5,6,7). Below are the comments: 

1. p.1 Abstract: U-NET may be an important term which should be included in the abstract. The proposed reverse time migration algorithm is also the proposal of the research. Please clearly state U-NET (CFFM+ESAM) followed by RTM is the proposal. 

2. p.7 Eq. 8 may be strange because numerator and denominator has the same component Rm. 

3. p.8 Fig. 6 (a) What kind of subsurface environment is represented in the model of fissures and uneven bedrock and concrete? Brief explanation is needed. 

4. p.10 Eq. 10 I did not understand what is x and y. If y is groundtruth images, how did you define groundtruth reflection patterns of radar images? 

5. p.12 fig.11 please show the schematics of buried pipes to indicate the corresponding reflection patterns in the radar images. The timescale of fig. 14 and depth range of fig. 15 may not be consistent. 

6. p.12 fig.11,fig.12,fig.14,fig.15 I am not sure about the improvement of the proposed method compared to res-unet. The echos may be important but not observed in the simulation. You may evaluate images by comparing traces of signals or indices you proposed. 

7. Related to q.6 above, please clarify the meaning of "fidelity" which is used many times in the manuscript. 

Reviewer 2 Report

What were some of the problems that GPR, or ground-penetrating radar, ran into as it was being made?
Which parts make it possible for the receptive field to be bigger while keeping the same resolution and number of parameters?
A refined spatial attention module (ESAM) has also been added to make sure that all important information is included in GPR recordings.The difficulty that led to the invention of its replacement, the hybrid loss function, was caused by classical loss functions. So, what precisely is the issue?
When employing ESAM, how many spatial attention modules does clutter removal account for?
What technologies have been used successfully to solve problems like finding crops, finding illnesses in buildings, finding cracks in the pavement, and researching glaciers?
Which of these features does CFFM use to obtain more accurate semantic segmentation?
Which part was made to make it easier for the network to pick up the specific data transmissions that the intended signal sent out?
What makes it less important to use context-relevant information when using different growth rates?
What kind of module is made when dilated convolution is used on a dataset to make multi-scale and multi-resolution feature maps?
With the frameworks that are already in place, which of the following deep learning methods might be the easiest to use? 

Comparison with recent studies and methods would be appreciated.

Introduction section can add the issues in the current work context and how proposed algorithms/approaches can overcome this.

Literature review techniques have to be strengthened by including the current system's issues and how the author proposes to overcome the same.

Clarify the finding Error rate and accuracy in the performance analysis section.

It is suggested to add the chart for the given process with a description.

The mapping process for the proposed technique should be discussed in detail.

Conclusion should state scope for future work.

Reviewer 3 Report

Article ID: remotesensing-2250442

Article Title: Deep Learning for Improved Subsurface Imaging: Enhancing GPR Clutter Removal Performance Using Contextual Feature Fusion and Enhanced Spatial Attention

In this paper, it proposed a novel deep learning framework for reconstructing GPR recordings that incorporates a contextual feature fusion module (CFFM) to expand the receptive field without losing resolution or increasing the number of parameters.Therefore, it is interesting and attractive. However, it should be major revised to enhance the quality, as follows:

1) In Section 1, authors should make three sub sections, motivation, contributions and organization of the paper

2) Literature review is not upto the mark. Pl include one table for comparative studies of the present research

3) A summary table should be provided for convenience for the readers in literature review section with comparison analysis of other approaches

4) Contributions of the research paper is limited, Pl at least three contributions should be there in any journal article

5) Overall work methodology.is not clear. Pl elaborate it clearly

6) Eq 7 is not clear. Pl elaborate  all the parameters  clearly .

7) Section 3.2 should derive more with respect to the present research

8)  Conclusion is not upto the mark. Pl avoid the numbering in conclusion section

9) Finally, the authors should double-check all formation, typos, and writing throughout the paper.

Round 2

Reviewer 1 Report

I appreciate the extensive response and modifications from the authors. About the measurement and validation part I understand the situations. The results may seem improved compared to the previous manuscript. Anyway, I appreciate the submitted version as it is. 

Reviewer 3 Report

Authors are  well addressed within its area of research.